# Prevalence of Internet Addiction and Gaming Disorders in Southeast Asia: A Meta-Analysis

**DOI:** 10.3390/ijerph17072582

**Published:** 2020-04-09

**Authors:** Doris X. Y. Chia, Charis W. L. Ng, Gomathinayagam Kandasami, Mavis Y. L. Seow, Carol C. Choo, Peter K. H. Chew, Cheng Lee, Melvyn W. B. Zhang

**Affiliations:** 1National Addictions Management Service, Institute of Mental Health, Singapore 539747, Singapore; doris_xy_chia@imh.com.sg (D.X.Y.C.); charis_wl_ng@imh.com.sg (C.W.L.N.); gomathinayagam_kandasami@imh.com.sg (G.K.); cheng_lee@imh.com.sg (C.L.); 2Department of Psychology, Institute of Mental Health, Singapore 539747, Singapore; mavis_seow@imh.com.sg; 3Department of Psychology, James Cook University, Singapore 387380, Singapore; 4Te Wānanga o Ngā Kete, Department of Arts, Law, Psychology & Social Sciences, University of Waikato, Hamilton 3240, New Zealand; ccychoo@gmail.com; 5College of Healthcare Sciences, Division of Tropical Health and Medicine, James Cook University, Townsville QLD 4811, Australia; peter.chew@jcu.edu.au; 6Family Medicine & Primary Care, Lee Kong Chian School of Medicine, Nanyang Technological University Singapore, Singapore 308232, Singapore

**Keywords:** Internet addiction, gaming disorders, Southeast Asia, prevalence, meta-analysis

## Abstract

This meta-analytic review aimed to examine the pooled prevalence rates of Internet addiction and gaming disorders in Southeast Asia. Several databases including PubMed, MEDLINE, PsycINFO, Web of Science, Embase, and Cochrane Central were searched and a total of 24 studies were included in this study. The selection of studies was conducted in accordance to the preferred reporting items for systematic reviews and meta-analyses (PRISMA) guidelines. Two meta-analyses were conducted to examine data on Internet addiction and gaming disorders separately. A random-effects model was employed to derive the pooled prevalence rate. Mixed-effects meta-regression and subgroup analyses were performed to examine the moderators of the between-study heterogeneity. Publication bias was tested using the Egger’s regression test and funnel plot. Only seven out of the 11 Southeast Asian countries were represented in the literature. All except for two of the included studies were cross-sectional in nature. The findings revealed a pooled prevalence rate of 20.0% (95% confidence interval: 14.5%–27.0%) and 10.1% (95% confidence interval: 7.3%–13.8%) for Internet addiction and gaming disorders respectively. Mean age and study population were significant moderators of the between-study heterogeneity in the prevalence rates of gaming disorders such that samples involving older participants showed higher prevalence rate than those involving younger individuals. Country of study was found to be significant moderator of the between-heterogeneity for both Internet addiction and gaming disorders, however the findings should be interpreted with caution due to the small and unbalanced sample sizes. There was no significant publication bias. Such epidemiology research should be extended to the Southeast Asian countries that have not been studied or are under-studied. Given that the prevalence rates appear to be higher in Southeast Asia than in other world regions, future research should also explore the factors behind these inter-regional differences. Further longitudinal studies should also be conducted to examine the trajectories of such disorders.

## 1. Introduction

Over the past few decades, technological breakthroughs have fundamentally shaped the role of the Internet in our lives. The Internet has now become an integral part of our daily living across school, work, and leisure. In Southeast Asia, the Internet Penetration Rate (IPR), defined as the percentage of the total population who use the Internet, was the highest for Brunei as of June 2019 (94.9%), followed by Singapore (84.5%) and Thailand (82.2%) [1]. 

This widespread use of the Internet has resulted in significant concerns with regards to problematic Internet behaviors and related conditions. Based on the Diagnostic and Statistical Manual of Mental Disorders, 4th Edition (DSM-IV) diagnostic criteria on Pathological Gambling, Young [2] has defined Internet addiction as the excessive use of Internet for leisure purposes over six months meeting five or more of the following criteria: preoccupation with the Internet; spending longer time online to achieve gratification; unsuccessful attempts to reduce or quit Internet use; experience of negative moods such as irritability associated with cut-down or quit attempts; using the Internet for longer periods than expected; negatively impacted school, work, or relationships; concealment about the extent of Internet use; and using the Internet to cope with problems or negative moods. Similarly, the American Psychiatric Association [3] has suggested for the inclusion of Internet Gaming Disorder (IGD) as a condition for further study in the latest edition of the Diagnostic and Statistical Manual of Mental Disorders (DSM-5). The DSM-5 outlines IGD as the presence of dysfunctional online gaming behavior that is marked by at least five of the proposed symptoms: preoccupation, psychological withdrawal, tolerance, failure to cut down or stop, loss of interest in previously enjoyable pursuits, constant gaming despite ramifications, lying about time spent on gaming, using gaming to cope with negative affect, and possible or actual negative repercussions on employment and social relationships due to gaming [3]. Whilst there have been suggested diagnostic criteria for both Internet addiction and gaming disorders, thus far, only gaming disorders have been officially recognized as a psychiatric disorder. The World Health Organization [4] has also formally included Gaming Disorder (GD) as a mental health condition in the recently updated International Classification of Diseases (ICD-11). The ICD-11 recognizes GD as dysfunctional gaming behavior, both online and offline, marked by inability to control, prioritization of gaming over other activities, and persistent or exacerbated gaming behavior despite negative life consequences [4]. 

Whilst there have been developments in the refinement of the diagnostic criteria for these disorders, there has since been more research done in this area. Notably, there has been a proliferation of epidemiological studies done to establish the prevalence rates of Internet and gaming disorders globally [5,6]. Given the high IPR in Southeast Asia, it is of vast importance to synthesize independent studies to achieve a holistic understanding on the prevalence rates of such disorders in the region. These rates will help facilitate policy planning and the allocation of resources both for the screening and the treatment of such disorders in the region. 

To the authors’ best of knowledge, there has only been one such study, Balhara and colleagues [7], that has attempted to conduct a narrative review in this area. However, whilst the aim of their review was to focus on Southeast Asian studies, only two out of the 29 epidemiological studies examined were conducted in the Southeast Asia region (Indonesia and Thailand). The majority of the included studies (22 out of 29) were conducted in India, which is a South Asian country. The Southeast Asian countries include Brunei, Myanmar, Cambodia, Timor-Leste, Indonesia, Laos, Malaysia, Philippines, Singapore, Thailand, and Vietnam. Hence, the prior review has not provided an accurate review of the rates of these disorders in Southeast Asia. In addition, the review was centered solely on studies conducted on student samples. 

In order to address this research gap, this current study aimed to examine the pooled prevalence rates of Internet addiction and gaming disorders in Southeast Asia through a meta-analytic review of the available epidemiological studies conducted in the region. Meta-analysis is a quantitative approach to literature synthesis through the use of statistics and significance testing [8,9]. As compared to traditional qualitative methods, such as narrative and systematic reviews which are prone to subjectivity due to human interpretation, meta-analysis allows for a more objective literature synthesis by quantifying outcomes [8,9]. This is especially useful for the synthesis of epidemiological studies, such as the current study, where a pooled estimate of the prevalence rates can be calculated. In addition, moderators driving the between-study heterogeneity can also be explored using the heterogeneity indices [10]. Past epidemiological studies on Internet and gaming disorders have primarily focused on single age group, with a lack of comparison across groups. As such, this study aimed to examine the moderating effect of age group, operationalized as population type (e.g., children, adolescents, adults, or mixed) and mean age, on the between-study heterogeneity. This study also examines gender as a moderator, as past studies have shown that males are more susceptible to such disorders [11,12]. Lastly, inter-regional differences were also explored using country as a moderator. As compared to Balhara et al. [7], the current study has a broader focus by taking into account studies conducted on general populations, inclusive of non-student samples.

## 2. Methods

### 2.1. Search Strategy

In order to identify the relevant articles, the following databases were searched from inception till 26th January 2020, that of PubMed, MEDLINE, PsycINFO, Web of Science, Embase, and Cochrane Central. The following search terminologies were used for MEDLINE, that of “epidemiology” or “prevalence” and “Internet” or “gaming” or “digital gaming” and “addict” or “disorder” or “problematic” or “pathological” or “excessive” or “dependence” or “addictive behaviour”. As the interest was on articles from Southeast Asian countries, the following additional search terminologies were applied, that of “Thai” or “Singapore” or “Malaysia” or “Indonesia” or “Brunei” or “Myanmar” or “Vietnam” or “Cambodia” or “Lao” or “Filipino” or “Philippines” or “Southeast Asia”. This MEDLINE search strategy was adapted for the rest of the databases. The search strategy for each of the databases have been included as a Appendix A. The authors have consulted a library information specialist in the conceptualization of the terminologies and key words used in the search strategy, and in the refinement of the search for the various databases.

### 2.2. Inclusion and Exclusion Criteria

Studies were included if they were: (1) original research, (2) examined prevalence rates of Internet or gaming disorders, (3) conducted in the Southeast Asia region, (4) conducted in general populations, and (5) the abstract was available in English language. Studies were excluded if they were: (1) non-original research (such as review studies), (2) did not report prevalence rates of Internet or gaming disorders, (3) not conducted within the Southeast Asia region, (4) conducted with psychiatric populations, or (5) the abstract was not available in English language.

### 2.3. Ethical Approval

As meta-analysis is considered to be a form of secondary research involving the extraction of findings from prior studies, therefore ethical approval was not required for the current study.

### 2.4. Selection of Studies

The selection of studies was conducted in accordance to the preferred reporting items for systematic reviews and meta-analyses (PRISMA) guidelines. The title and abstract of the studies identified from the databases using the aforementioned search strategy were first screened for relevancy to the study topic on Internet and gaming disorders by two independent authors. The full-text article of the remaining studies were then assessed against the inclusion and exclusion criteria to determine the final pool of studies to be included in the current meta-analysis.

### 2.5. Data Extraction

Data from the included studies on Internet addiction and gaming disorders were extracted separately into two standardized spreadsheets by author DC. The following data were extracted: (1) citation including last name of first author and year of publication, (2) country in which the study was conducted, (3) study design, (4) study population, (5) sample size, (6) characteristics of sample including gender breakdown and mean age, (7) assessment tool utilized, and (8) prevalence rate. To ensure data accuracy, the extracted data were cross-checked by a second author (MZ). Any disagreements were resolved by means of a discussion between the authors DC and MZ. If they were still not able to resolve the disagreements, a third author was consulted.

### 2.6. Data Analyses

All statistical analyses were conducted using the Comprehensive Meta-Analysis Version 2.0 Program. The extracted data on Internet addiction and gaming disorders were analyzed separately via two independent meta-analyses. For the purpose of meta-analysis, the extracted data were coded into formats that were compatible with the program, e.g., categorical variables in text format were coded into numerical categories and percentages were represented in decimal format.

For each of the two conditions, a random-effects model was employed to derive the pooled estimate of the prevalence rates, based on 95% confidence interval. This type of model assumes that the included studies are a subset of all possible relevant studies, accounting for true heterogeneity in effect sizes across studies on top of systematic sampling error, allowing for the generalization of findings beyond the context of the included studies [10,13]. A forest plot was also generated to provide a visual representation of the prevalence data from the included studies.

Heterogeneity across studies was assessed through significance testing of the I^2^ statistic, with null hypothesis assuming homogeneity. Based on the heuristic recommended in Higgins et al. [14], an I^2^ of 25%, 50%, and 75% reflect low, moderate, and high levels of heterogeneity respectively.

Following the significant between-studies heterogeneity found, mixed-effects meta-regression analyses using unrestricted maximum likelihood estimation were conducted to ascertain if the continuous moderators, including mean age and proportion of male participants, were significantly associated with the between-study heterogeneity in prevalence rates. For meta-regression, the regression coefficients and the corresponding *Z* and *p* values were reported. For categorical moderators, including country and study population, sub-group analyses using a mixed-effects model were done.

A funnel plot was generated to assess publication bias, with asymmetry suggesting possible presence of publication bias. Egger’s regression test was also conducted to examine the significance of publication bias, with *p* < 0.05 signifying the presence of a significant publication bias. If a significant publication bias was shown, a Classic Fail-Safe test would be conducted to ascertain the number of omitted studies needed to counter the publication bias.

## 3. Results

An initial pool of 184 citations were identified from the databases based on the search strategy. A total of 27 articles were found to be duplicates and were excluded. The screening of study titles and abstracts for relevancy to study topic resulted in the exclusion of 110 citations. Another 23 studies were subsequently excluded after the assessment of full-text article against the inclusion and exclusion criteria: did not report prevalence data on Internet addiction or gaming disorders (12 studies), non-original study (one study), use of psychiatric sample (one study), no access to full-text article (six studies), and sharing of duplicated dataset with an included study (three studies). A final count of 24 studies were included for the current study (refer to Figure 1). The aforementioned steps on the selection of studies are detailed in Figure 1.

Of the included studies, eight studies were conducted in Malaysia, six in Singapore, five in Thailand, three in Vietnam, one study examined data from Malaysia and Philippines, and the remaining study looked at data from Indonesia, Malaysia, Myanmar, Thailand, and Vietnam. In terms of the condition studied, 16 studies were conducted on Internet addiction, six studies on Internet gaming disorder or gaming disorders, and two studies looked at both Internet addiction and gaming disorders. The study populations included adults (12 studies), adolescents (six studies), children (one study), children and adolescents (two studies), and adolescents and adults (three studies). All except for two studies were cross-sectional in nature.

For studies that examined more than one condition or country, the data on each condition or country were considered as unique data points for analysis. The characteristics of the included studies on Internet addiction and gaming disorders, including study details, country, study design, population type, sample size, mean age, gender breakdown, assessment tool, and prevalence rates, can be found in Table 1 and Table 2, respectively.

### 3.1. Internet Addiction

From the meta-analysis based on a random-effects model, the pooled prevalence rate of Internet addiction was found to be 20.0% (95% confidence interval: 14.5%–27.0%, *Z* = −6.955, df = 22, *τ*^2^ = 0.883, *I*^2^ = 98.745). The forest plot shown in Figure 2 illustrates the prevalence rates found in the individual studies.

There was significant between-study heterogeneity, *p* < 0.001. Based on the heuristic recommended by Higgins et al. [14], the *I*^2^ of 98.7% represented high level of heterogeneity. The mixed-effects meta-regression revealed that both mean age (ß = −0.084, *Z* = −0.746, *p* = 0.456) and proportion of male participants (ß = 7.024, *Z* = 1.835, *p* = 0.067) were not significant moderators of the between-study heterogeneity. The meta-regression findings are summarized in Table 3. The mixed-effects subgroup analyses showed that only country of study was a significant categorical moderator, with Thailand having the highest point estimate (44.7%), followed by Indonesia (38.5%), Vietnam (21.6%), Malaysia (19.2%), Myanmar (16.1%), Singapore (9.4%), and Philippines (4.9%). On the other hand, study population was not found to be a significant moderator. The findings on the subgroup analyses can be found in Table 4.

There was no significant publication bias as indicated by the non-significant Egger’ regression test (intercept = −4.358, 95% confidence interval = −13.7%–5.0%, *t* = 0.968, df = 21, *p* = 0.344).

### 3.2. Gaming Disorders

The pooled prevalence rate of gaming disorders based on a random-effects model was found to be 10.1% (95% confidence interval: 7.3%–13.8%, *Z* = −11.899, df = 7, *τ*^2^ = 0.253, *I*^2^ = 97.4). The forest plot of prevalence rates of gaming disorders is illustrated in Figure 3.

There was significant between-study heterogeneity, *p* < 0.001. According to the heuristic suggested by Higgins et al. [14], the *I*^2^ of 97.4% represented a high level of heterogeneity. The mixed-effects meta-regression revealed that mean age (ß = 0.065, *Z* = 9.250, *p* = 0.000) was a significant moderator of the between-study heterogeneity, while proportion of male participants was not significant (ß = 0.027, *Z* = 0.023, *p* = 0.981). The meta-regression findings are shown in Table 5. The mixed-effects subgroup analyses showed that both the country of study (ß = 0.069, *Z* = −34.074, *p* = 0.000) and study population (ß = 0.102, *Z* = −66.852, *p* = 0.000) were significant categorical moderators. The findings on subgroup analyses are detailed in Table 6. In particular, the studies conducted in Singapore showed higher pooled prevalence estimate of gaming disorders (13.0%) than of Thailand (5.7%). In terms of study population, the pooled prevalence estimate was the highest for adolescents and adults (17.7%), followed by adults (15.4%), children and adolescents (9.3%), children (7.5%), and adolescents (5.4%).

There was no significant publication bias as indicated by the non-significant Egger’ regression test (intercept = 2.437, 95% confidence interval = −12.7%–17.6%, *t* = 0.393, df = 6, *p* = 0.708).

## 4. Discussion

In terms of the IPR as of January 2020, the Southeast Asia region was ranked 9th (66.0%), with the first being Northern Europe (95.0%), followed by Western Europe (92.0%), and Northern America (88.0%) [37]. Despite the lower IPR, the pooled prevalence rates of Internet addiction (20.0%) and gaming disorders (10.1%) for the Southeast Asia region found in the current study are substantially higher than what have been found for other world regions. In their meta-analysis of 80 studies with mean participant age of 18.42 years old, Cheng et al. [38] found the global prevalence of Internet addiction to be 6.0% while the prevalence rates for the seven world regions examined were: Middle East (10.9%), North America (8.0%), Asia (7.1%), South and East Europe (6.1%), Oceania (4.3%), North and West Europe (2.6%), and South America (0.0%). In terms of gaming, Müller et al. [39] examined the prevalence of Internet gaming disorder among seven European countries using data from 12,938 adolescents aged 14 to 17 and found the overall prevalence rate to be 1.6%, with the highest rate found for Greece (2.5%), followed by Poland (2.0%), Iceland (1.8%), Germany (1.6%), Romania (1.3%), The Netherlands (1.0%), and Spain (0.6%). Similarly, in their study of 1178 adolescents aged 8 to 18 in the United States, Gentile [40] found the prevalence rate of pathological videogaming to be 8.5%. Cultural differences may offer a possible explanation for the higher prevalence rates observed in Southeast Asia despite the lower IPR. Western countries, such as the American and European countries, are often perceived as having a more individualistic culture than the Southeast Asian countries [41,42]. Interestingly, Arpaci et al. [43] found that a higher level of individualism, manifested by increased needs for achievement, autonomy, and dominance, and reduced need for affiliation, was significantly associated with a lower degree of Internet addiction. Accordingly, this higher level of individualism may have accounted for the lower prevalence of Internet and gaming disorders in Western countries than in Southeast Asia.

As indicated by the significant tests of heterogeneity and *I*^2^ of 98.7% and 97.4%, the current meta-analysis revealed high levels of between-study heterogeneity in the prevalence rates of Internet addiction and gaming disorders. The subsequent meta-regression and subgroup analyses explored mean age, proportion of male participants, country, and study population as potential moderators that may have contributed to the heterogeneity. In particular, mean age was found to be a significant moderator for gaming disorders, such that higher mean age of sample was associated with higher prevalence rate. Older individuals, such as college students and working adults, often have more autonomy and opportunities to utilize smart devices. This, coupled with their technological proficiency and the lack of external supervision, may put them at higher risk for gaming disorders. On the other hand, parents of younger children or adolescents, particularly in the Asian culture, may impose supervision and restrictions on their children’s involvement in gaming behaviors due to academic concerns, reducing their risk for pathological gaming [44,45]. Besides the lack of external supervision, psychological motivations may also explain higher levels of gaming among older individuals. According to Erikson’s Theory of Psychosocial Development [46], adolescents will actively explore and establish their self-identity at the “Identity versus Role Confusion” stage while adults will seek to build up their social relationships at the “Intimacy versus Isolation” stage. These intrapersonal and interpersonal needs are often shown to motivate gaming behaviors, particularly the involvement in the massively multiplayer online role-playing games [47,48]. In their study with 179 Korean undergraduates, Kim and Kim [47] found that social motivations significantly predicted excessive Internet gaming. In particular, the study showed that participants who were more motivated to overcome loneliness, build real-life social network, or increase virtual social network, were more likely to engage in excessive Internet gaming. In another study with 509 young adults in Croatia, it was revealed that lower levels of self-concept were associated with higher levels of problematic Internet gaming [48]. As such, adolescents and adults may be more inclined than children to play virtual games to build their self-identity and social relationships or to compensate such unfulfilled needs in real life. In a similar trend, the current study also revealed that study population was a significant moderator, such that samples from older populations including adolescents and adults (17.7%) and adults only (15.4%) showed higher prevalence rates of gaming disorders than samples from younger populations including children and adolescents (9.3%), children (7.5%), and adolescents (5.4%). Interestingly, such trends were not observed for Internet addiction, such that both mean age and study population were not found to be significant moderators. This disparity in findings may be indicative of the distinctiveness and specificity of gaming disorders in the broader context of Internet addiction, which encompasses a wide range of other behaviors including gambling, social media, and shopping. However, these findings on the moderating effect of population type should be considered preliminary as the sample sizes of studies for each level of the moderator were small and unbalanced for Internet Addiction and gaming disorders respectively (adults (*n* = 16; *n* = 2), adolescents (*n* = 5; *n* = 2), children (*n* = 0; *n* = 1), adolescents and adults (*n* = 2; *n* = 1), and children and adolescents (*n* = 0; *n* = 2)). For both conditions, country of study was shown to be a significant moderator. Likewise, this finding should be interpreted with caution as not all Southeast Asian countries were represented, and for the represented countries, there was only limited and unbalanced number of studies. For instance, only seven out of the 11 Southeast Asian countries were represented in the epidemiological literature for Internet addiction (Malaysia (*n* = 10), Vietnam (*n* = 4), Singapore (*n* = 3), Thailand (*n* = 3), Indonesia (*n* = 1), Myanmar (*n* = 1), and Philippines (*n* = 1)). Similarly, only two countries were represented for gaming disorders (Singapore (*n* = 5) and Thailand (*n* = 3)).

Given the high levels of heterogeneity, there may be other moderators that were not explored in the current study due to the lack of data. For instance, the heterogeneity may be accounted for by the diversity in the assessment tools employed by the included studies i.e., varied assessment tools and varied cut-off criterions. A total of 10 different assessment tools were employed across the 23 studies on Internet addiction and five were used across the eight studies on gaming disorders. The varied assessment tools were observed to produce differing prevalence rates. Using the Internet Addiction Test (IAT) and the Revised Chen Internet Addiction Scale (CIAS-R), Mak et al. [19] examined the prevalence of Internet addiction in several countries, including Malaysia and Philippines, and found substantially different rates across the two scales. The study showed prevalence rates of 2.4% (IAT) and 14.1% (CIAS-R) for Malaysia, and 4.9% (IAT) and 21.1% (CIAS-R) for Philippines. Additionally, the use of varied cut-off criterions for the same scale across studies conducted on similar samples have also resulted in differing prevalence rates. In their studies to examine the prevalence of Internet addiction among college students in Singapore, Tang et al. [26,27] employed the 12-item IAT with a cut-off of 40 points in one study and 36 points in the other to identify participants with Internet addiction. Despite the similarities in sampling, the two studies revealed differing rates of 4.9% and 9.3% respectively.

As with all studies, the current study has its own limitations. Firstly, despite having the aim of synthesizing the epidemiological studies on Internet and gaming disorders in Southeast Asia, not all Southeast Asian countries were represented due to the availability of literature. Moreover, the number of studies for each country was limited. The lack of studies may affect the generalizability of findings to Southeast Asia as a whole. Next, given the high levels of between-study heterogeneity, there may be other potential moderators that were not examined. As mentioned above, the assessment tools utilized in the included studies were diversified in terms of differing scales and cut-off criterions. However, these potential moderators were left unexplored as subgroup and meta-regression analyses were not feasible due to the limited number of studies. Additionally, the subgroup analyses on the moderating effects of population type and country on the between-study heterogeneity should be considered to be exploratory due to the small and unbalanced sample sizes of studies to represent each level of the moderators. In addition, the prevalence rate in one study, Ng et al. [20], did not correlate with the raw data provided in the table summarizing the characteristics of the participants, and another study (Boonvisudhi et al. [17]) did not report the gender breakdown based on the full sample which was included as a moderator in the current study. We have attempted to reach out to the authors via email for further clarifications, but to date, the authors have not responded. For the purpose of our current meta-analysis, we have recomputed the prevalence using the raw data results instead of the reported prevalence for Ng et al. [20] and excluded Boonvisudhi et al. [17] for the meta-regression which used proportion of male participants as the moderator. Lastly, given that MEDLINE and Scopus often have overlaps in journal indexing, only MEDLINE but not Scopus was searched for this current study. As such, there may be relevant studies not included in the current study as the journals in which they are published in are only indexed on Scopus. However, the authors note that such possibility is low due to the similarity in journal indexing across the two databases.

Despite the limitations, the current study is one of first efforts to synthesize the epidemiological literature on Internet and gaming disorders in the Southeast Asia context. Compared to traditional qualitative reviews, the study conducted meta-analysis which is a more robust and objective method to literature synthesis. By using meta-analysis, the study was able to derive pooled estimates as proxies of the prevalence of Internet and gaming disorders in Southeast Asia. Moreover, the study has also explored potential moderators that may influence the prevalence rates.

Through the current findings, several future research directions have been identified. To the best of the authors’ knowledge, the epidemiology of Internet and gaming disorders have not been examined in some Southeast Asia countries including Brunei, Cambodia, Timor-Leste, and Laos. For those countries that have been covered, most were underrepresented. Accordingly, more research work can be done in the Southeast Asia countries to better understand such disorders in the region. Next, the study has revealed higher prevalence rates in Southeast Asia as compared to other world regions. Future studies should seek to explore the contributing factors and nuances behind these inter-regional differences. Lastly, all except for two of the included studies were cross-sectional in nature, future research may seek to conduct more longitudinal studies to examine the trajectories of Internet and gaming disorders.

There are several clinical implications that arise from the current findings. The prevalence of these disorders implies there being a need to screen for these disorders proactively amongst individuals who are coming forth for treatment. Screening should also be done amongst individuals with other psychiatric disorders, such as attention deficit hyperactivity disorder, social anxiety disorder, depressive disorder and alcohol abuse, as prior studies have reported there being an association between Internet Addiction, gaming disorders and these other psychiatric comorbidities [49,50].

## 5. Conclusions

In conclusion, the current meta-analysis found the pooled prevalence rates of Internet addiction and gaming disorders in Southeast Asia to be 20.0% and 10.1% respectively. The meta-regression and subgroup analyses showed that mean age and study population were significant moderators of the between-study heterogeneity in prevalence rates of gaming disorders with older samples showing higher prevalence rates than younger samples. Even though country of study was found to be a significant moderator for both Internet addiction and gaming disorders, findings should be interpreted with caution due to the small and unbalanced sample sizes. Future studies should extend such epidemiological research to Southeast Asian countries that have not been studied or under-studied. Given that the prevalence rates of Internet addiction and gaming disorders in Southeast Asia appear to be higher than that in other world regions, future research should examine the factors behind these inter-regional differences. Further longitudinal studies should also be conducted to examine the trajectories of such disorders.

## Figures and Tables

**Figure 1 ijerph-17-02582-f001:**
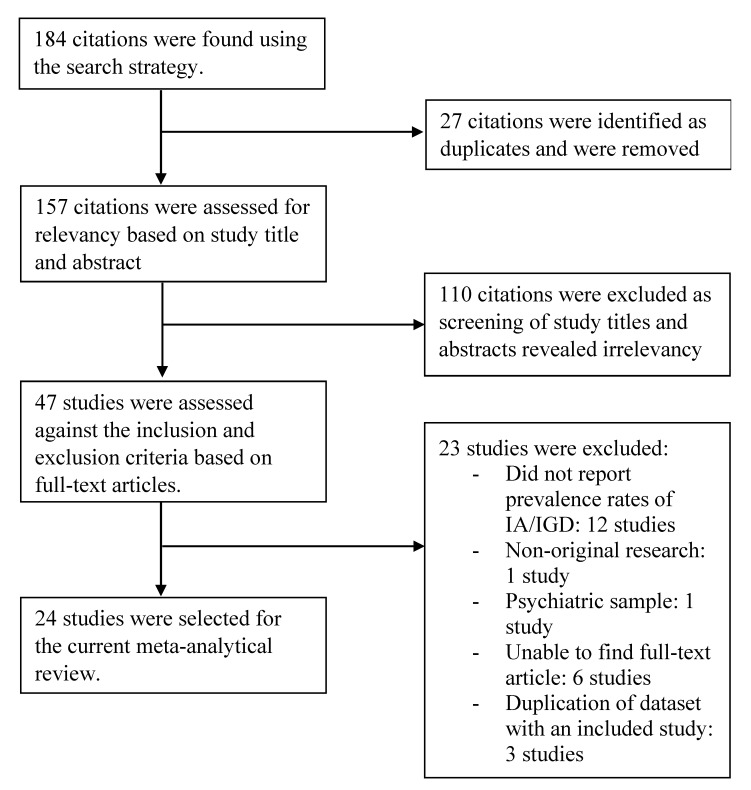
Flowchart depicting the process of study selection.

**Figure 2 ijerph-17-02582-f002:**
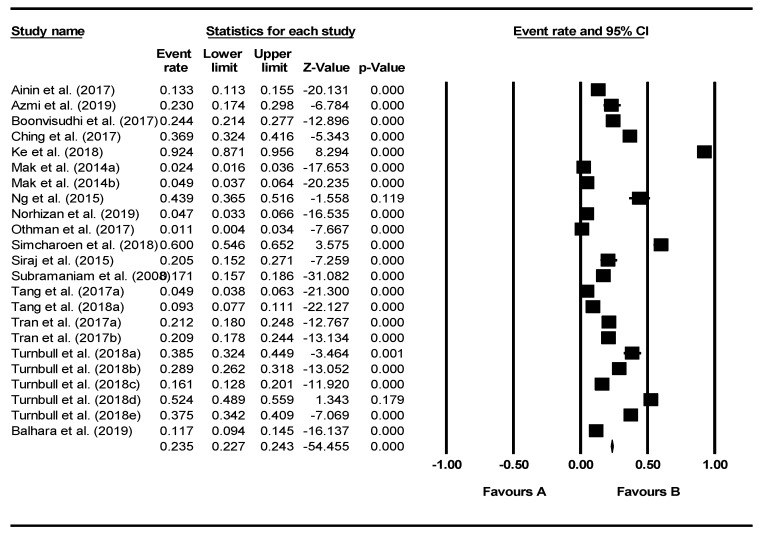
Forest plot of prevalence rates of Internet addiction.

**Figure 3 ijerph-17-02582-f003:**
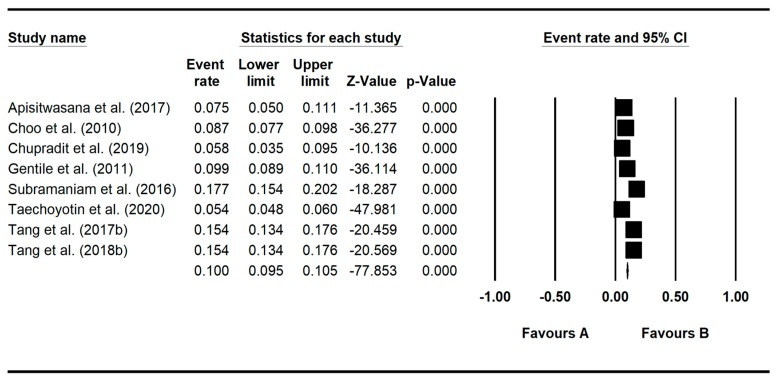
Forest plot of prevalence rates of gaming disorders.

**Table 1 ijerph-17-02582-t001:** Characteristics of the included studies on Internet addiction.

Paper	Country	Study Design	Population	Sample Size	Sample Characteristics	Assessment Tool	Prevalence
Ainin et al. (2017) [15]	Malaysia	Cross-sectional	Adults	1000	50.3% males and 49.7% females Mean age = N.A. years old (SD = N.A.)	20-item Young’s Internet Addiction Test (IAT)	13.3%
Azmi et al. (2019) [16]	Malaysia	Cross-sectional	Adolescents	178	36.5% males and 63.5% females Mean age = 10.6 years old (SD = 1.57)	Malay Validated Internet Addiction Test (MVIAT)	23.0%
Boonvisudhi et al. (2017) [17]	Thailand	Cross-sectional	Adults	705	N.A.% males and N.A.% females Mean age = 20.51 years old (SD = 1.91)	Young Diagnostic Questionnaire (YDQ)	24.4%
Ching et al. (2017) [11]	Malaysia	Cross-sectional	Adults	426	36.6% males and 63.4% females Mean age = 21.6 years old (SD = 1.5)	MVIAT	36.9%
Ke et al. (2018) [18]	Malaysia	Longitudinal	Adolescents	157	46.0% males and 54.0% females Mean age = 14.0 years old (SD = N.A.)	Problematic Internet Use Questionnaire (PIUQ)	92.4% (Time 1)
Mak et al. (2014a) [19]	Malaysia	Cross-sectional	Adolescents	969	46.0% males and 54.0% females Mean age = 14.7 years old (SD = 1.2)	IAT	2.4%
Mak et al. (2014b) [19]	Philippines	Cross-sectional	Adolescents	999	38.5% males and 61.5% females Mean age = 16.4 years old (SD = 1.7)	IAT	4.9%
Ng et al. (2015) [20]	Malaysia	Cross-sectional	Adults	164	34.1% males and 65.9% females Mean age = 19.02 years old (SD = 0.19)	YDQ	43.9%
Norhizan et al. (2019) [21]	Malaysia	Cross-sectional	Adults	674	23.0% males and 77.0% females Mean age = 21.66 years old (SD = 1.89)	IAT	4.7%
Othman et al. (2017) [22]	Malaysia	Cross-sectional	Adults	267	13.9% males and 86.1% females Mean age = 20.9 years old (SD = 1.4)	MVIAT	1.1%
Simcharoen et al. (2018) [23]	Thailand	Cross-sectional	Adults	324	43.2% males and 56.8% females Mean age = 20.88 years old (SD = 1.81)	Thai version of the IAT	0.6%
Siraj et al. (2015) [24]	Malaysia	Cross-sectional	Adults	176	26.7% males and 73.3% females Mean age = N.A. years old (SD = N.A.)	Internet Addiction Diagnostic Questionnaire (IADQ)	20.5%
Subramaniam et al. (2008) [25]	Singapore	Cross-sectional	Adolescents	2735	49.3% males and 50.6% females Mean age = 13.9 years old (SD = 1.0)	>5 h spent on the Internet per day	17.1%
Tang et al. (2017a) [26]	Singapore	Cross-sectional	Adults	1107	37.4% males and 62.6% females Mean age = 21.45 years old (SD = 1.80)	12-item Young’s IAT	4.9%
Tang et al. (2018a) [27]	Singapore	Cross-sectional	Adults	1119	38.0% males and 61.9% females Mean age = 21.52 years old (SD = 1.89)	12-item Young’s IAT	9.3%
Tran et al. (2017a) [28]	Vietnam	Cross-sectional	Adolescents and Adults	566	38.9% males and 61.1% females Mean age = 21.5 years old (SD = 3.8)	Vietnamese version of the 12-item IAT	21.2%
Tran et al. (2017b) [29]	Vietnam	Cross-sectional	Adolescents and Adults	589	36.8% males and 63.2% females Mean age = 21.7 years old (SD = 1.7)	Vietnamese version of the 12-item IAT	20.9%
Turnbull et al. (2018a) [30]	Indonesia	Cross-sectional	Adults	231	N.A. % males and N.A. % females Mean age = N.A. years old (SD = N.A.)	YDQ	38.5%
Turnbull et al. (2018b) [30]	Malaysia	Cross-sectional	Adults	1023	N.A. % males and N.A. % females Mean age = N.A. years old (SD = N.A.)	YDQ	28.9%
Turnbull et al. (2018c) [30]	Myanmar	Cross-sectional	Adults	386	N.A. % males and N.A. % females Mean age = N.A. years old (SD = N.A.)	YDQ	16.1%
Turnbull et al. (2018d) [30]	Thailand	Cross-sectional	Adults	783	N.A. % males and N.A. % females Mean age = N.A. years old (SD = N.A.)	YDQ	52.4%
Turnbull et al. (2018e) [30]	Vietnam	Cross-sectional	Adults	817	N.A. % males and N.A. % females Mean age = N.A. years old (SD = N.A.)	YDQ	37.5%
Balhara et al. (2019) [31]	Vietnam	Cross-sectional	Adults	617	28.0% males and 72.0% females Mean age = 21.0 years old (SD = 2.3)	Generalized Problematic Internet Use Scale-2 (GPIUS2)	11.7%

N.A.: Not Available; SD: Standard Deviation.

**Table 2 ijerph-17-02582-t002:** Characteristics of the included studies for Internet Gaming Disorder/Gaming Disorder.

Paper	Country	Study Design	Population	Sample Size	Sample Characteristics	Assessment Tool	Prevalence
Apisitwasana et al. (2017) [32]	Thailand	Cross-sectional	Children	295	52.9% males and 47.1% females Mean age = 9.87 years old (SD = 0.7)	Game Addiction Screening Test (GAST)	7.5%
Choo et al. (2010) [12]	Singapore	Cross-sectional	Children and Adolescents	2998	72.7% males and 27.3% females Mean age = 11.2 years old (SD = 2.06)	10-item screening tool based on DSM-IV Pathological Gambling	8.7%
Chupradit et al. (2019) [33]	Thailand	Cross-sectional	Adolescents	242	33.5% males and 66.5% females Mean age = N.A. years old (SD = N.A.)	GAST	5.8%
Gentile et al. (2011) [34]	Singapore	Longitudinal	Children and Adolescents	2998	72.7% males and 27.3% females Mean age = N.A. years old (SD = N.A.)	10-item screening tool based on DSM-IV Pathological Gambling	9.9% (Time 1)
Subramaniam et al. (2016) [35]	Singapore	Cross-sectional	Adolescents and Adults	972	63.2% males and 36.8% females Mean age = 23.6 years old (SD = 5.0)	Internet Gaming Disorder Questionnaire (IGDQ)	17.7%
Taechoyotin et al. (2020) [36]	Thailand	Cross-sectional	Adolescents	5497	48.1% males and 37.6% females Mean age = N.A. years old (SD = N.A.)	Thai Version of the Internet gaming disorder test (IGD-20 Test)	5.4%
Tang et al. (2017b) [26]	Singapore	Cross-sectional	Adults	1107	37.4% males and 62.6% females Mean age = 21.45 years old (SD = 1.80)	12-item Problematic Online Gaming Questionnaire	15.4%
Tang et al. (2018b) [27]	Singapore	Cross-sectional	Adults	1119	38.0% males and 61.9% females Mean age = 21.52 years old (SD = 1.89)	12-item Problematic Online Gaming Questionnaire	15.4%

**Table 3 ijerph-17-02582-t003:** Meta-regression of mean age and proportion of male participants on prevalence rates of Internet addiction.

Moderators	Number of Studies Used	Slope	Standard Error	95% CI: Lower Limit	95% CI: Upper Limit	*Z*	*p*
Mean age	16	−0.084	0.112	−0.303	0.136	−0.746	0.456
Proportion of male participants	17	7.024	3.828	−0.479	14.527	1.835	0.067

CI: Confidence Interval.

**Table 4 ijerph-17-02582-t004:** Subgroup analyses on the effect of country and population the prevalence rates of Internet addiction.

Subgroups	No. of Studies Used	Pooled Prevalence (%)	95% CI	*p* Value for between-Group Comparison
Indonesia	1	38.5	32.4–44.9	
Malaysia	10	19.2	10.6–32.3	
Myanmar	1	16.1	12.8–20.1	
Philippines	1	4.9	3.7–6.4	
Singapore	3	9.4	4.5–18.5	
Thailand	3	44.7	24.8–66.5	
Vietnam	4	21.6	12.8–34.1	0.000 *
Overall:	23	17.0	15–19.1	
Adults	16	20.1	13.6–28.8	
Adolescents	5	19.6	6.6–45.4	
Adolescents and Adults	2	21.0	18.8–23.5	
Overall:	23	21.0	18.8–23.3	0.966

CI: Confidence Interval. *p* < 0.001 *.

**Table 5 ijerph-17-02582-t005:** Meta-regression of mean age and proportion of male participants on prevalence rates of gaming disorders.

Moderators	Number of Studies Used	Slope	Standard Error	95% CI: Lower Limit	95% CI: Upper Limit	*Z*	*p*
Mean age	5	0.065	0.007	0.051	0.079	9.250	0.000 *
Proportion of male participants	8	0.027	1.144	−2.216	2.269	0.023	0.981

CI: Confidence Interval. *p* < 0.001 *.

**Table 6 ijerph-17-02582-t006:** Subgroup analyses on the effect of country and population the prevalence rates of gaming disorders.

Subgroups	No. of Studies Used	Pooled Prevalence (%)	95% CI	*p* Value for between-Group Comparison
Singapore	5	13.0	9.8–17.0	
Thailand	3	5.7	4.8–6.7	
Overall:	8	6.9	6.0–7.9	0.000 *
Adults	2	15.4	14.0–17.0	
Adolescents	2	5.4	4.9–6.0	
Children	1	7.5	5.0–11.1	
Children and Adolescents	2	9.3	8.2–10.5	
Adolescents and Adults	1	17.7	15.4–20.2	
Overall:	8	10.2	9.6–10.8	0.000*

CI: Confidence Interval. *p* < 0.001 *.

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
