# Peer review of "Prevalence of Internet Addiction and Gaming Disorders in Southeast Asia: A Meta-Analysis"

_ijerph, 2020, doi:10.3390/ijerph17072582_

Round 1

Reviewer 1 Report

In the manuscript entitled “A Meta-analytical Review of the Prevalence of Internet 2 Addiction and Gaming Disorders in Southeast Asia”, the authors aimed to analyze the pooled prevalence rates of Internet addiction and Gaming disorders in Southeast Asia, conducting two meta-analyses (separately for Internet addiction and gaming disorders).

The text is overall clear and well-written. However, there are some points that require improvement.

In particular, the following minor concerns should be addressed in order to improve the manuscript:

- In the introduction, authors describe the characteristics of Gaming Disorders but do not report any definition of Internet Addiction. In my opinion, a broader definition of Internet Addiction is needed (more could have been said on this controversial concept).

-  The authors used mean age as potential moderator. Could the Authors better explain such choice?

- Lines 241- 245. The authors state: “the current study also revealed that study population was a significant moderator such that samples from older populations including adolescents and adults (17.7%) and adults only (15.4%) showed higher prevalence rates of gaming disorders than samples from younger populations including children and adolescents (9.3%), 244 children (7.5%), and adolescents (5.4%).” Considering that there is only one study with a population of children, the authors should underline this aspect among the limitations of the study.

-According to the authors, the lack of external supervision could explain that the higher mean age was associated with a higher prevalence rate of gaming disorders. In my opinion, this is not the only reason. The authors should provide a possible interpretation of this data considering the “psychological motivations” that may influence the Internet Gaming Disorders

Author Response

Reviewer 1’s Comments

We would like to thank Reviewer 1 for your positive comments and your valuable suggestions to improve on our manuscript. Please refer to our point-by-point replies below.

In the manuscript entitled “A Meta-analytical Review of the Prevalence of Internet 2 Addiction and Gaming Disorders in Southeast Asia”, the authors aimed to analyze the pooled prevalence rates of Internet addiction and Gaming disorders in Southeast Asia, conducting two meta-analyses (separately for Internet addiction and gaming disorders).

The text is overall clear and well-written. However, there are some points that require improvement.

In particular, the following minor concerns should be addressed in order to improve the manuscript:

- In the introduction, authors describe the characteristics of Gaming Disorders but do not report any definition of Internet Addiction. In my opinion, a broader definition of Internet Addiction is needed (more could have been said on this controversial concept).

We agree with Reviewer 1 that it is important to include the definition of Internet Addiction in the context of this study. Given that Internet Addiction has not been formally included in DSM-5 or ICD-11, we have incorporated the definition by Dr Kimberly Young who is a well-known subject matter expert in the topic of Internet disorders. The definition of Internet Addiction has been added to revised manuscript (Lines 51 to 58).

-  The authors used mean age as potential moderator. Could the Authors better explain such choice?

We have added in the rationale for the selection of the moderators from Lines 95 to 100 in the Introduction. The amends are as follows:

“Past epidemiological studies on Internet and gaming disorders have primarily focused on single age group, with a lack of comparison across groups. As such, this study aimed to examine the moderating effect of age group, operationalized as population type (e.g., children, adolescents, adults or mixed) and mean age, on the between-study heterogeneity. This study has also looked at gender as a moderator as past studies have shown that males are more susceptible to such disorders [11, 12]. Lastly, inter-regional differences were also explored using country as a moderator”

We have considered the use of mean age as a potential moderator, given the fact that most of our studies have reported the mean ages of their sampled participants. Thus, we have decided to investigate if the mean age of the included sample have affected the heterogeneity of the computed pooled prevalence.

- Lines 241- 245. The authors state: “the current study also revealed that study population was a significant moderator such that samples from older populations including adolescents and adults (17.7%) and adults only (15.4%) showed higher prevalence rates of gaming disorders than samples from younger populations including children and adolescents (9.3%), 244 children (7.5%), and adolescents (5.4%).” Considering that there is only one study with a population of children, the authors should underline this aspect among the limitations of the study.

We added in Lines 281 to 284 under the Discussion to acknowledge the small sample sizes of studies as a study limitation and have reiterated this point under the Limitations section from Lines 315 to 318.

-According to the authors, the lack of external supervision could explain that the higher mean age was associated with a higher prevalence rate of gaming disorders. In my opinion, this is not the only reason. The authors should provide a possible interpretation of this data considering the “psychological motivations” that may influence the Internet Gaming Disorders

We have incorporated psychological motivations in terms of the fulfillment of intrapersonal and interpersonal needs as an alternative explanation for the higher levels of gaming among older individuals (Lines 259 to 272).

Reviewer 2 Report

This study is more quantitative, but not qualitative evaluation of internet addiction and gaming disorder.

Results section  - please describe more detail what findings and calculations presented in tables and figures. This is a lack of information about the main knowledge which you want to tell.

Discussion. As it is more related to quantitative evaluation, discussion section must be orientated to statistical methods, why you choose such type of evaluation, what this evaluation shows - you must discuss about results. I like discussion section as qualitative evaluation - it is fine. But discussion section in your work must be from two parts: evaluation methods and qualitative (which is perfect).

Author Response

Reviewer 2’s Comments

We would like to thank Reviewer 2 for highlighting these important points. Please see our point-by-point replies below.

This study is more quantitative, but not qualitative evaluation of internet addiction and gaming disorder.

We like to clarify that this study is a meta-analytical study and we have performed a quantitative analysis.

Results section - please describe more detail what findings and calculations presented in tables and figures. This is a lack of information about the main knowledge which you want to tell.

The following lines have been added to briefly describe the information that can be found in the figures and tables- Lines 176 to 177, 187 to 190, 194 to 195, 200, 204 to 205, 210 to 211, 216, and 218 to 219.

Discussion. As it is more related to quantitative evaluation, discussion section must be orientated to statistical methods, why you choose such type of evaluation, what this evaluation shows - you must discuss about results. I like discussion section as qualitative evaluation - it is fine. But discussion section in your work must be from two parts: evaluation methods and qualitative (which is perfect).

We seek to clarify that the statistical methods that were used have been comprehensively explained under data-analyses in the methods. Please find as enclosed our methods:

“All statistical analyses were conducted using the Comprehensive Meta-Analysis Version 2.0 Program. The extracted data on Internet addiction and gaming disorders were analyzed separately via two independent meta-analyses. For the purpose of meta-analysis, the extracted data were coded into formats that were compatible with the program, e.g., categorical variables in text format were coded into numerical categories and percentages were represented in decimal format.

For each of the two conditions, a random-effects model was employed to derive the pooled estimate of the prevalence rates, based on 95% confidence interval. This type of model assumes that the included studies are a subset of all possible relevant studies, accounting for true heterogeneity in effect sizes across studies on top of systematic sampling error, allowing for the generalization of findings beyond the context of the included studies [10,13]. A forest plot was also generated to provide a visual representation of the prevalence data from the included studies.

Heterogeneity across studies was assessed through significance testing of the I2 statistic, with null hypothesis assuming homogeneity. Based on the heuristic recommended in Higgins et al. [14], an I2 of 25%, 50%, and 75% reflect low, moderate, and high levels of heterogeneity respectively.

Following the significant between-studies heterogeneity found, mixed-effects meta-regression analyses using Unrestricted Maximum Likelihood estimation were conducted to ascertain if the continuous moderators, including mean age and proportion of male participants, were significantly associated with the between-study heterogeneity in prevalence rates. For meta-regression, the regression coefficients, corresponding Z and P values were reported. For categorical moderators, including country and study population, sub-group analyses using a mixed-effects model were done.

A funnel plot was generated to assess publication bias, with asymmetry suggesting possible presence of publication bias. Egger’s regression test was also conducted to examine the significance of publication bias, with p < .05 signifying presence of significant publication bias. If significant publication bias was shown, a Classic Fail-Safe Test would be conducted to ascertain the number of omitted studies needed to counter the publication bias.”

The methods that we have presented in this paper are in-line with prior published meta-analyses, such as (Zhang MWB, Lim RBC, Lee C, Ho RCM., Prevalence of Internet Addiction in Medical Students: a Meta-analysis. Acad Psychiatry. 2018 Feb;42(1):88-93. doi: 10.1007/s40596-017-0794-1. Epub 2017 Aug 28. Review.) and Zhang MW, Ho RC, Cheung MW, Fu E, Mak A., Prevalence of depressive symptoms in patients with chronic obstructive pulmonary disease: a systematic review, meta-analysis and meta-regression. Gen Hosp Psychiatry. 2011 May-Jun;33(3):217-23. doi: 10.1016/j.genhosppsych.2011.03.009. Epub 2011 Apr 27. Review.

Reviewer 3 Report

The paper is interesting, and it is discussing two important social and actual topics. However, I noted some issues that the Authors should clarify:
1) They used several databases but not Scopus. Why?
2) I think could be better to explicit the exact period related to the selected studies and why
3) In the text they use the expression meta-analysis, but in the title no
4) I think could be better to mention the PRISMA approach to conduct the study
5) In the first part, the literature review, the Authors should introduce the moderator role of the variables that they identified in the results

Author Response

Reviewer 3’s Comments

We would like to thank Reviewer 3 for your positive comments and useful recommendations. Please see our point-by-point replies below.

The paper is interesting, and it is discussing two important social and actual topics. However, I noted some issues that the Authors should clarify:

1) They used several databases but not Scopus. Why?

For the purposes of this current meta-analysis, we have decided to focus on major medical databases, namely that of PubMed, MEDLINE and PsycINFO. Since these disorders, that of Internet addiction and Internet gaming are relatively new, we anticipate that most of the published papers will be indexed on these databases. Also, given the similarity of journal indexing across MEDLINE and Scopus, only MEDLINE was considered for this current study. We acknowledge that this may be a potential limitation and have incorporated it under the Limitations section from Lines 325 to 329.

2) I think could be better to explicit the exact period related to the selected studies and why

The databases were searched from their inception till 26th Jan 2020, the date when we began our search on the databases. Lines 105 to 107 have been amended to better reflect this.

3) In the text they use the expression meta-analysis, but in the title no

The title has been changed to, “Prevalence of Internet Addiction and Gaming Disorders in Southeast Asia: A Meta-analysis”.

4) I think could be better to mention the PRISMA approach to conduct the study

We added in Lines 22 to 23 in the Abstract and Line 128 in the Methods Section to indicate that the selection of studies was conducted in accordance to the PRISMA guidelines.

5) In the first part, the literature review, the Authors should introduce the moderator role of the variables that they identified in the results

We have added in the rationale for the selection of the moderators from Lines 95 to 100 in the Introduction.